# The Effects of Sodium Acetate on the Immune Functions of Peripheral Mononuclear Cells and Polymorphonuclear Granulocytes in Postpartum Dairy Cows

**DOI:** 10.3390/ani13172721

**Published:** 2023-08-26

**Authors:** Cong Yuan, Dejin Tan, Zitong Meng, Maocheng Jiang, Miao Lin, Guoqi Zhao, Kang Zhan

**Affiliations:** 1Institute of Animal Culture Collection and Application, College of Animal Science and Technology, Yangzhou University, Yangzhou 225009, China; yuancongyzu@163.com (C.Y.); zulutango7@163.com (Z.M.); jmcheng1993@163.com (M.J.); linmiao@yzu.edu.cn (M.L.); gqzhao@yzu.edu.cn (G.Z.); 2Institutes of Agricultural Science and Technology Development, Yangzhou University, Yangzhou 225009, China; 3Joint International Research Laboratory of Agriculture and Agri-Product Safety, The Ministry of Education of China, Yangzhou University, Yangzhou 225009, China

**Keywords:** acetate, peripheral blood mononuclear cell, polymorphonuclear granulocyte, postpartum

## Abstract

**Simple Summary:**

There is little information about the effect of sodium acetate (NaAc) on alleviating pro-inflammation and oxidative stress and enhancing the immune function of peripheral mononuclear cells (PBMCs) and polymorphonuclear granulocytes (PMNs) in postpartum dairy cows. Our results demonstrated that supplementation of NaAc in postpartum dairy cows exhibited positive roles in healthy dairy cows, reflective of antimicrobial and adhesive function enhancement in PBMCs and PMNs. In conclusion, our study provided a novel resolution strategy in which the use of NaAc enhances the antimicrobial and adhesive abilities of PBMCs and PMNs and improves immunity in postpartum dairy cows.

**Abstract:**

Excessive lipid mobilization will snatch cell membrane lipids in postpartum dairy cows, which may impair the function of immune cells, including peripheral mononuclear cells (PBMCs) and polymorphonuclear granulocytes (PMNs). Acetate, as a precursor and the energy source of milk fat synthesis, plays a key role in lipid synthesis and the energy supply of dairy cows. However, there is little information about the effect of sodium acetate (NaAc) on the immune function of PBMC and PMN in postpartum dairy cows. Therefore, this study aimed to evaluate the effects of NaAc on the immune functions of PBMCs and PMNs in postpartum dairy cows. In this experiment, twenty-four postpartum multiparous Holstein cows were randomly selected and divided into a NaAc treatment group and a control group. Our results demonstrated that the dietary addition of NaAc increased (*p* < 0.05) the number of monocytes and the monocyte ratio, suggesting that these postpartum cows fed with NaAc may have better immunity. These expressions of genes (*LAP*, *XBP1*, and *TAP*) involved in the antimicrobial activity in PBMCs were elevated (*p* < 0.05), suggesting that postpartum dairy cows supplemented with NaAc had the ability of antimicrobial activity. In addition, the mRNA expression of the monocarboxylate transporters *MCT1* and *MCT4* in PBMCs was increased (*p* < 0.05) in diets supplemented with NaAc in comparison to the control. Notably, the expression of the *XBP1* gene related to antimicrobial activity in PMN was upregulated with the addition of NaAc. The mRNA expression of genes (*TLN1*, *ITGB2*, and *SELL*) involved in adhesion was profoundly increased (*p* < 0.05) in the NaAc groups. In conclusion, our study provided a novel resolution strategy in which the use of NaAc can contribute to immunity in postpartum dairy cows by enhancing the ability of antimicrobial and adhesion in PBMCs and PMNs.

## 1. Introduction

The perinatal period in dairy cows is defined as the time interval from day 21 antepartum to day 21 postpartum [1], and it is one of the most special periods in the physiological stages of dairy cows suffering from many challenges in terms of physiology, metabolism, and feeding [2]. Extensive nutritional, metabolic, and hormonal changes during the postpartum period in dairy cows are risk factors for both metabolic and infectious diseases [3,4]. Postpartum dairy cows experience a negative energy balance (NEB) due to decreased dry matter intake (DMI) and increased energy requirements for milk production [5]. During the NEB period, there is a high incidence of various diseases, including ketosis, metritis, mastitis, and so on. It is reported that the occurrence of various diseases is likely related to the suppression of the immune function in postpartum dairy cows [6], which is attributed to the disruption of the function of the immune cells.

Peripheral blood mononuclear cells (PBMCs) are the precursor of most immune cells, and polymorphonuclear granulocytes (PMNs) are the first line of defense against invading pathogens, especially those causing inflammation and oxidative stress [7]. Both of them are considered to be risk factors contributing to increased susceptibility to disease for postpartum dairy cows [8]. In addition, inflammation caused by metabolic disorders plays a vital role in the induction of oxidative stress in postpartum dairy cows [9]. There is evidence suggesting that high oxidative stress leads to a decrease in the milk production of dairy cows [10] and substantial economic losses. Therefore, postpartum dairy cows are characterized by impaired immune function, pro-inflammation, and high oxidative stress, which may be mitigated by the recovery of PBMC and PMN function. PBMC is the most commonly used cell model in immunology research, which can reflect the occurrence of infection and disease in dairy cows. The antioxidant stress ability of PBMCs is closely related to the antioxidant and immune function of dairy cows [11]. PMNs reversibly adhere to the endothelium extracellular matrix and initiate the chemotactic process [12]. The chemotactic function and phagocytosis of PMNs are weakened when immunosuppression occurs in the postpartum period. The morphology and function of PMNs undergo dramatic changes during the growth process, and the size of the cell and the shape of the nucleus also change greatly, forming the proteins necessary for PMNs to play the role of phagocytosis and sterilization [13]. However, excessive adipose mobilization in postpartum dairy cows, resulting from an NEB, is responsible for the degree of inclusion of fatty acids in the leukocyte membrane and disrupts the membrane function of immune cells, which is well-known for being involved in impairing immune functions and dysfunctional inflammation [14]. Thus, dietary supplementation of additives related to lipid precursors contributes to fat synthesis and reduces dysregulated lipid mobilization and loss of lipids involved in the leukocyte membrane synthesis, which may restore the function of immune cells and increase antibacterial function in postpartum dairy cows.

Acetate, which is the main product of rumen fermentation, maintains a high concentration in peripheral blood and is the raw material for the de novo synthesis of fatty acids [15]. It is used to synthesize fatty acids for the utilization of adipose and mammary tissue and satisfies the energy demands of various tissues. Acetate has shown the ability to reduce endoplasmic reticulum (ER) stress proteins in bovine mammary epithelial cells without compromising cell viability [16], which makes it a promising option for reducing ER stress and ensuring optimal milk production in cows. In addition, acetate exhibits antimicrobial activities against most mastitis pathogens compared to other acids [17]. Acetate increases the stress resistance of dairy cows through its antibacterial ability, which reduces the energy loss of dairy cows during the synthesis of immunoglobulins or resistance to pathogenic microorganisms. As a result, dairy cows become qualified to redistribute energy, protein, and lipids for milk synthesis. Supplementation with sodium acetate (NaAc) has been shown to increase milk fat yield and concentration [18], enhance rumen digestibility, and improve DMI [19], indicating a beneficial role of acetate in improving the performance of dairy cows. Excessive lipid mobilization in postpartum dairy cows will seize cell membrane lipids, which may impair the function of immune cells, including PBMCs and PMNs. Supplementation with acetate can promote de novo synthesis of lipids. However, there is little information about the effect of NaAc on alleviating the pro-inflammation and oxidative stress and enhancing the immune function of PBMCs and PMNs in postpartum dairy cows.

We hypothesized that supplementation of acetate could increase the immune function of PBMCs and PMNs owing to a decrease in dysregulated lipid mobilization in the postpartum period, thereby improving the immunity of postpartum dairy cows. Therefore, the aim of this study was to evaluate the effects of NaAc on the immune functions of PBMCs and PMNs in postpartum dairy cows, which provides a scientific theoretical basis for the regulation of acetate in the production of dairy cows.

## 2. Materials and Methods

### 2.1. Animals

The following experimental procedures were conducted in accordance with the principles of the Institutional Animal Care and Use Committee of Yangzhou University (SYXK (Su) IACUC 2012-0029). Experimental period was 21 d in total. Twenty-four multiparous Holstein cows (0 d postpartum) were randomly classified into two treatment groups: control (diet without NaAc supplementation, *n* = 12) or 8 mol sodium acetate/d (656 g/d) (diet with NaAc supplementation, *n* = 12). These cows were housed in a tie-stall barn located at Yangzhou University Dairy Production Research and Teaching Center. All dairy cows were fed once daily at 08:00 h at roughly 110% of the expected intake to satisfy 100% of NRC demands (Table 1), and refusals to feed were carefully registered.

### 2.2. Blood Collection and Sampling Analysis

After 3 h of the morning feeding on the last day of treatments, the 10 mL whole blood samples were collected from tail veins using disposable blood collection needles and vacuum sampling vessels with negative pressure. A portion of the samples was packed into evacuated tubes containing Ethylene Diamine Tetraacetic Acid (EDTA) and was immediately sent to the Animal Hospital of Yangzhou University (Yangzhou, China) for routine blood testing. Red blood cell count (RBC), hematocrit (HCT), hemoglobins (HGBs), mean corpuscular volume (MCV), mean corpuscular hemoglobin (MCH), mean corpuscular hemoglobin concentration (MCHC), red blood cell distribution width (RDW), white blood cell (WBC) count, neutrophili granulocytes (NEUs), lymphocytes (LYMs), eosinophils (EOS), monocytes (MONOs), and basophils (BASOs) were measured by automatic animal blood analyzer (Mindray; BC-240 Vet). The remaining tubes were left standing for 30 min and then centrifuged at 3000 r/min for 20 min to produce serum samples, which were stored at −80 °C.

### 2.3. Isolation of Peripheral Blood Mononuclear Cells

A total of 5 mL of peripheral blood was transferred to heparin tubes, 3 mL of Lympholyte was added to the bottom of the 15 mL sterile tubes, and then equal volumes of blood and PBS (2 mL each) were gently added to the Lympholyte surface. These were centrifuged at 18 °C and 2000 r/min for 30 min. The centrifuged lymphocyte (2–3 mL) was transferred to a new 15 mL sterile tube, followed by PBS to 15 mL. Then, cells were filtered with a 40 µm filter, and the filtered cells were transferred to a 50 mL pipette into a new sterile tube. After that, these were centrifuged at 2000 r/min and 4 °C for 10 min, and the supernatant was drained. Then, 800 µL of Lysing Buffer was added to suspend the cells, allowing the Lysing Buffer to be added to the total volume of 4 mL and left for 5 min at room temperature. These cells were centrifuged at 4 °C and 2000 r/min for 10 min, and the supernatant was drained. After removing the supernatant, 800 µL PBS was added to suspend the cells, and then phosphate-buffered saline (PBS) was added to the total volume of 15 mL. Repeat the centrifugal operation of the previous step. Finally, 800 µL of suspended cells was added and transferred to a 1 mL enzyme-free tube, centrifuged at 12,000/min for 1 min, and then the supernatant was discarded and transferred to a liquid nitrogen tank for preservation.

### 2.4. Isolation of Polymorphonuclear Granulocytes

The PMNs were isolated from whole blood samples using a method modified by Garcia M et al. [20]. The venous blood was transferred to 15 mL sterile tubes containing the 2.2 mL anticoagulant citrate dextrose (ACD-A), which was gently mixed by inversion and placed on ice. These tubes were centrifuged at 1000× *g* at 4 °C for 20 min, and the plasma, buffy coat, and one-third of the red blood cells were removed. The remaining cells were transferred to 50 mL sterile tubes, and 18 mL of ice-cold deionized water was added to lyse the red blood cells, enabling the tubes to be gently reversed for no more than 45 s. Then, the 2 mL 10 × PBS solution was immediately added to the tubes to restore the isotonic condition, and then tubes were centrifuged at 1000× *g* at 4 °C for 10 min to remove the supernatant. The remaining precipitates were washed with 20 mL of calcium- and magnesium-free Hank’s balanced salt solution (CMF-HBSS) and centrifuged at 850× *g* at 4 °C for 5 min. Eventually, the isolated PMNs were cultured in Roswell Park Memorial Institute (RPMI-1640) medium, in which 5% inactivated fetal bovine serum (FBS) and 4 mmol/L glutamine were added.

### 2.5. Determination of Related Gene Expression

Total RNA extraction kit (Tiangen Biochemical Technology., Ltd., Beijing, China) was used to extract RNA from PBMCs and PMNs, and the specific operation steps were carried out according to the instructions of the manual. The concentration and purity of the total RNA were measured using a One Drop^TM^ ultramicro spectrophotometer. Reverse transcription reagents were used in the reverse transcription of RNA into cDNA. The specific operation is as follows: The total volume of reverse transcription is 10 µL. A total of 2 µL of 5 × PrimeScript RT Master Mix was added, and then a certain amount of RNA-free enzyme water was absorbed and added to ensure that the total RNA content was 500 ng. This was mixed with a gun, and then a finger tube was placed into a PCR machine for reaction. Reaction conditions: 37 °C, 15 min, and 85 °C for 5 s. Primer 6.0 was used to design primers, and the primer sequence was submitted to Genewiz Inc. for synthesis. The primer sequence is shown in Table 2. The target gene was quantified by LightCycler^®^96. The PCR reaction system was 20 µL; the system is shown in Table 3. The PCR conditions were set as follows: 1 cycle at 95 °C for 30 s, 40 cycles at 95 °C for 5 s, and 60 °C for 30 s, followed by a melting curve program (60 to 95 °C). The GAPDH, β-actin, and RPS9 internal reference genes were used for correction. The relative gene expression was calculated by 2^−ΔΔct^.

### 2.6. Statistical Analysis

These results were plotted via distribution of normality and analyzed using the one-way ANOVA program in SPSS 25.0 (SPSS Inc.; Chicago, IL, USA). Duncan’s method was used for multiple comparisons; the results were tested by the independent sample test, and the results were expressed as mean. Differences were declared significant at *p* < 0.05, and tendencies were significant at 0.05 < *p* < 0.10.

## 3. Results

### 3.1. Effects of NaAc on Blood Routine Indexes in Postpartum Dairy Cows

Compared with the control group, the NaAc group saw a profound effect (*p* < 0.05) on the number of monocytes and the monocyte ratio (Table 4), and the cows remained healthy without infection and mastitis during the whole experiment, suggesting that these cows fed with NaAc may have better immunity. This may be attributed to the effect NaAc has on reducing the dysregulated lipid mobilization in postpartum dairy cows.

### 3.2. Gene Expression of Peripheral Blood Mononuclear Cells

The expression of the functional genes in PBMCs is presented in Table 5. The expression of genes encoding in ER stress (*GPR78*, *ATF4*, *sXBP1*, *EIF2A*, and *HSP70*) was not affected by NaAc supplementation relative to the control group. However, *ASK1* tended to be elevated by the supplementation of NaAc, suggesting that the ER stress function of PBMCs may be affected by the supplementation of NaAc. Remarkably, compared with the control group, the mRNA expression of genes involved in antimicrobial activity (*LAP*, *XBP1*, and *TAP*) was significantly increased (*p* < 0.05) by the dietary supplementation of NaAc, indicating that dietary supplementation of NaAc can effectively enhance the antimicrobial activity of PBMCs and resist the invasion of pathogenic bacteria. In addition, the mRNA expression of the *ICAM1* gene related to the adhesion molecule also tended to be upregulated (*p* = 0.07), suggesting that PBMCs can interact with immune cells and promote an immune response to enhance the antimicrobial ability resulting from the addition of NaAc. At the same time, the mRNA expression of *SOD2* genes involved in anti-oxidative stress was markedly increased (*p* < 0.05) in the NaAc group relative to control, demonstrating that NaAc may have a positive effect on regulating the ability of anti-oxidative stress of PBMCs. Notably, the mRNA expression of monocarboxylate transporter *MCT1* and *MCT4* was increased in the diet supplemented with NaAc in comparison to the control. It is concluded that supplementation of NaAc can promote the NaAc transport into PBMCs to synthesize fatty acids for the utilization of the membrane assembly.

### 3.3. Gene Expression of Polymorphonuclear Granulocyte

The expression of the functional genes in PMNs is presented in Table 6. The expression of genes encoded in short-chain fatty acid transports genes (*MCT1*, *MCT4*, and *GPR41*), those involved in oxidative stress (*NFE2L2*, *SOD2*, *NOX1*, and *CCL2*), and genes related to apoptosis and survival (*FAS*, *CASP2*, and *BAK1*) was not changed by NaAc supplementation compared with the control group. For the antimicrobial gene, the expression of *XBP1* was significantly upregulated with NaAc addition, suggesting that dietary addition of NaAc could significantly enhance the antimicrobial activity of PMNs. Remarkably, compared with the control group, the mRNA expression of genes involved in adhesion (*TLN1*, *ITGB2*, and *SELL*) was profoundly increased (*p* < 0.05) in the NaAc groups. It was strongly proven that NaAc can mount PMN immune responses to form a line of defense against the invasion of pathogens. In summary, NaAc supplementation can promote the function of PMNs by improving their adhesive, phagocytic, chemotactic, and antimicrobial functions.

## 4. Discussion

Previous studies have shown that an NEB exists in postpartum dairy cows, which will trigger body disorders, leading to the occurrence of oxidative stress, inflammation, and immunosuppression and eventually causing damage to the health of dairy cows. This increases breeding costs and causes economic losses [21]. In order to compensate for the lack of energy, the body will mobilize lipids for decomposition; however, excessive lipid mobilization triggers a mass of free fatty acids to accumulate in the liver, resulting in fatty liver in postpartum dairy cows, with triglyceride accumulation impairing liver function and increasing pro-inflammatory responses [22]. In addition, excessive lipid mobilization also despoils the amount of inclusion of the leukocyte membrane, triggering the disruption of the membrane function of PBMC and PMN cells [23]. Additionally, impaired PBMC and PMN functions also increase susceptibility to disease in postpartum dairy cows [8]. There are increasing results showing that the susceptibility of postpartum dairy cows to diseases is related to metabolic disorders and immunosuppression [24,25]. Therefore, it is crucial that a lipid precursor is found to promote fat synthesis and relieve the NEB in postpartum dairy cows, repair the function of immune cells, and enhance the immune system. As a precursor to the substance and energy source of milk fat synthesis, acetate plays a key role in lipid synthesis and the energy supply of dairy cows [26]. Previous studies pointed out that the appropriate amount of acetate was 5–10 mol/d [18,27]. According to the rumen volume and normal rumen acetate concentration of dairy cows, we finally determined that the supplemental amount of acetic acid was 8 mol/d (656 g/d). We speculated that dietary supplements with NaAc may play a key role in improving the functions of PBMCs and PMNs and increasing immunity in postpartum dairy cows.

PBMCs exist in all parts of the dairy cow’s body alongside its blood circulation and have the ability to reflect the overall condition of the dairy cow’s body [28]. Previous studies showed that immune responsiveness decreases gradually in the prepartum period and reaches its minimum expression immediately before calving [29,30]. The PBMCs are precursor cells for macrophagocytes, and they effectively regulate the inflammatory process by producing inflammatory mediators, such as cytokines and chemokines, to recruit other immune cells to phagocytize pathogenic bacteria [31]. *ICAM1* is a crucial gene related to the adhesion of PBMC [32]. *ICAM1* gene can promote the adhesion of inflammatory sites and regulate the body’s immune response. It plays an important role in stabilizing cell–cell interactions and promoting the migration of WBCs and endothelial cells. NaAc increased the mRNA expression of *ICAM1* in PBMCs, which interacts with another immune cell to phagocytize the pathogens. The expression of *LAP*, *XBP1*, and *TAP* was significantly increased; among them, *TAP* and *LAP*, as defensins, can provide innate defense against bacterial infection in dairy cows [33]. In fact, the postpartum period in dairy cows lasted for 21 days, and the upregulation of these genes involved in the antimicrobial activity is a sign of adaptation to a higher challenge with pathogens. In addition, the alternative hypothesis may be in line with the increased expression of SOD2. The amphiphilic and cationic properties of defensins lead to direct antimicrobial activity by disrupting bacterial membranes [34]. Meanwhile, defensins also contribute to host defense as immunomodulatory molecules, for example, by stimulating chemotaxis [35]. SOD2 is widely distributed in mitochondria and can effectively remove superoxide anions produced by the body, thereby reducing the peroxide damage. *MCT1* and *MCT4* perform the role of transporting short-chain fatty acids, and their gene family is responsible for the transport of monocarboxylic compounds across the plasma membranes. In the present study, the expression of genes involved in adhesion and antimicrobial activity in PBMCs was upregulated in the postpartum dairy cows that received NaAc supplementation, indicating that supplementation of NaAc can mount PBMC immune responses and improve the PBMC immunity in postpartum dairy cows.

PMNs, such as anti-inflammatory agents, are the primary line of defense in resisting invading pathogens [36]. Analysis of gene expression in the PMNs of postpartum cows revealed profound changes that affected the immune status of transitional cows [37]. Thomas and Schroder [38] considered that PMN pattern recognition receptor-mediated signaling is crucial for orchestrating innate and adaptive immunity via cytokine and chemokine production and antimicrobial release. As a key *XBP1* gene involved in Toll-like receptors, its signal pathway plays an important role in antimicrobial activity. Consequently, a higher expression of *XBP1* is thought to be beneficial for postpartum dairy cows. A variety of genes take part in the adhesion of PMNs. In the presence of infection, circulating neutrophils migrate to the site of infection and roll across, adhere to, and cross the endothelial barrier [39]. Thus, adhesion molecules play a crucial role in the recruitment of PMNs to the site of infection and the subsequent immune response [40]. Cell migration is mediated by a protein transcribed by the selectin gene *SELL*, which reduces the rolling rate of endothelial cells [41]. This gene product is essential for leukocyte binding and facilitating their movement on endothelial cells, thereby promoting cell migration to sites of inflammation. The interface integrin domain is interlinked with the cytoskeleton, and this binding of integrin to actin is mediated by *TLN1* [42]. *ITGB2*, which is mainly present on the surface of various white blood cells, is a crucial factor in the recruitment of PMNs to the site of inflammation, which can regulate cell adhesion and migration. In the adhesion of PMNs, we observed a significant increase in the expression of three genes (*TLN1*, *ITGB2*, and *SELL*) by supplementation with NaAc, suggesting that NaAc may regulate the functions of PMNs via enhancing adhesion stimulation of PMN activation [43]. This is notable, given that PMNs’ trafficking, phagocytosis, and pathogen-killing capacities are often impaired in the postpartum period [44], and changes in cytokines, metabolites, and hormone levels in the microenvironment for PMNs during this period are closely related to variations in the functional performance of PMN [45,46]. In the current study, a diet supplemented with NaAc can upregulate the mRNA expression of *XBP1*, *TLN1*, *ITGB2*, and *SELL*, demonstrating that the addition of NaAc can contribute to adhesive and antimicrobial functions, which exerts a beneficial role in neutralizing invading pathogens.

## 5. Conclusions

Our present study demonstrated that supplementation of NaAc in postpartum dairy cows exhibited positive roles in healthy dairy cows, reflective of antimicrobial and adhesive function enhancement in PBMCs and PMNs. These findings may be attributed to how NaAc contributes to fatty acids synthesis and reduction in dysregulated lipid mobilization and reduces the loss of lipids involved in the membrane synthesis of PBMCs and PMNs. Our study provided a novel resolution strategy where the use of NaAc can enhance the antimicrobial and adhesion ability of PBMCs and PMNs and improve immunity in postpartum dairy cows.

## Figures and Tables

**Table 1 animals-13-02721-t001:** Basal diet formulations and nutritional contents (% of DM).

Item	%
Alfalfa	3.76
Oat	12.83
Whole corn silage	39.39
Corn	14.18
Soya bean meal	18.91
Cotton seed meal	1.75
DDGS	4.8
Premix ^1^	4.38
Total	100
Nutrient levels ^2^	
CP	16.07
EE	3.84
NDF	33.44
ADF	24.42
Ash	7.28
Ca	0.74
P	0.48
NEL, MJ/kg ^3^	3.25

^1^ Vitamin and mineral mix contained 32 mg/kg Cu, 62.5 mg/kg Zn, 55 mg/kg Mn, 0.55 mg/kg Se, 75 mg/kg Fe, 1 mg/kg I, 4 mg/kg, 1.1 mg/kg vitamin A, 2.7 mg/kg vitamin D, 2 mg/kg vitamin E, and 4.2 mg/kg vitamin K_3_. ^2^ Measured values. ^3^ Predicted value of the NRC (2001).

**Table 2 animals-13-02721-t002:** Primers of target and housekeeping genes.

Gene	NCBI Accession No.	Sequence	Amplicon Length (bp)
*MCT1*	XM_015463657.2	F: CAATGCCACCAGCAGTTG	376
		R: GCAAGCCCAAGACCTCCAAT	
*MCT4*	NM_001109980.3	F: AGCGTCTGAGCCCAGGGAGG	223
		R: ACCTCGCGGCTTGGCTTCAC	
*GPR41*	XM_015458060.2	F: AACCTCACCCTCTCGGATCT	214
		R: GCCGAGTCTTGTACCAAAGC	
*GPR78*	NM_001075148.1	F: CGACCCCTGACGAAAGACAA	198
		R: AGGTGTCAGGCGATTTTGGT	
*ATF4*	XM_024991552.1	F: AGATGACCTGGAAACCATGC	190
		R: AGGGGGAAGAGGTTGAAAGA	
*sXBP1*	NM_001271737.1	F: TGCTGAGTCCGCAGCAGGTG	169
		R: GCTGGCAGACTCTGGGGAAG	
*NFE2L2*	XM_005202311.4	F: AAGGGACAAGTTGGAGCTGTT	145
		R: AATCCATGTCCCTTGACAGCAG	
*EIF2A*	XM_005211941.4	F: TCGTCATGTTGCTGAGGTCT	111
		R: GCACCATATCCGGGTCTCTT	
*SOD2*	NM_201527.2	F: GAGAAGGGTGATGTTACAGCTCAGA	100
		R: GGCTCAGATTTGTCCAGAAGATG	
*NOX1*	XM_024988030.1	F: GATCTGCAGGGAGATGGGTG	147
		R: GCTGCATGACCAGCAAAGTT	
*FAS*	NM_174662.2	F: AATGCCCACATGGCTGGTAT	131
		R: TTTTTCCGTTTGCCAGGAGG	
*CASP2*	XM_005205863.3	F: TGCTCCAGCTACAAGAGGTTTT	140
		R: AGCAGTGAACAGAAGGAGGTG	
*BAK1*	XM_024983575.1	F: CCAGAACCTAGCAGCACCAT	176
		R: ATACCGCTCTCAAACAGGCT	
*SELL*	NM_174182.1	F: CTCTGCTACACAGCTTCTTGTAAACC	104
		R: CCGTAGTACCCCAAATCACAGTT	
*ITGB2*	NM_175781.1	F: CCAGGTTATTCTATGGGCTCATG	102
		R: CCATACAAAATGTAGGCAATTCCTT	
*ICAM1*	NM_174348.2	F: AGAATTAGCGCTGACCTCTGTTAAG	100
		R: CGGACACATCTCAGTGACTAAACAA	
*TLN1*	NM_001205428.1	F: TTCCTGCCCAAGGAGTATGTG	100
		R: AGCGTACCTTGGCCTCAATCT	
*ASK1*	NM_001144081.2	F: GCTATGGAAAGGCAGCAGA	160
		R: TCTGCTGACATGGACTCTGG	
*MMP9*	NM_174744.2	F: CCCGGATCAAGGATACAGCC	177
		R: GGGCGAGGACCATACAGATG	
*LAP*	NM_203435.4	F: TGTCTGCTGGGTCAGGATTTAC	131
		R:TACTTGGGCTCCGAGACAGG	
*XBP1*	NM_001034727	F: CCGGAAGAAAGCTCGAATG	96
		R: TCTCGTAAAACGTGATTTTCTAACAA	
*TAP*	NM_174776.1	F: GAGGCTCCATCACCTGCTC	88
		R: GCTTACAGGATTTCCTACTCCTTG	
*NFKB1*	NM_001076409.1	F: TTCAACCGGAGATGCCACTAC	95
		R: ACACACGTAACGGAAACGAAATC	
*HSP70*	NM_203322.3	F: GTGCAGGAGGCGGAAAAGTA	183
		R: GGAAATCACCTCCTGGCACT	
*ALOX5*	NM_001192792.2	F: GAGAGATGGGCAAGCGAAGT	114
		R: GGGTTCCACTCCATCCATCG	
*CCL2*	NM_174006.2	F: GCTCGCTCAGCCAGATGCAA	117
		R: GGACACTTGCTGCTGGTGACTC	
Housekeeping genes			
*GAPDH*	NM_001034034.2	F: TTGTCTCCTGCGACTTCAACA	103
		R:TCGTACCAGGAAATGAGCTTGAC	
*β-actin*	AY141970.1	F: GACCCAGATCATGTTCGAGA	145
		R: CTCATAGATGGGCACCGTGT	
*RPS9*	NM_001101152.2	F: CCTCGACCAAGAGCTGAAG	64
		R: CTCATAGATGGGCACCGTGT	

Abbreviations: MCT1, monocarboxylate transporter 1; MCT4, monocarboxylate transporter 4; GPR41, G-protein coupled receptor 41; GPR78, G-protein coupled receptor 78; ATF4, activating transcription factor 4; sXBP1, X-box- binding protein-1; NFE2L2, nuclear factor erythroid 2-related factor 2; EIF2A, eukaryotic translation initiation factor 2A; SOD2: superoxide dismutase 2; NOX1, mitogenic oxidase 1; FAS, tumor necrosis factor receptor superfamily member 6; CASP2, Caspase-2; BAK1, Bcl-2 homologous antagonist/killer; SELL, L-selectin; ITGB2, integrin beta-2; ICAM1, intercellular adhesion molecule 1; TLN1, Talin-1; ASK1, apoptotic signal-regulating kinase 1; MMP9, matrix metalloproteinase-9; LAP, lingual antimicrobial peptide; XBP1, X-box-binding protein 1; TAP, yracheal antimicrobial peptide; NFKB1, nuclear factor NF-kappa-B p105 subunit; HSP70, heat shock protein 70; ALOX5, Polyunsaturated fatty acid 5-lipoxygenase; CCL2, C-C motif chemokine 2; GAPDH, Glyceraldehyde-3-phosphate dehydrogenase; and RSP9, flagellar radial spoke protein 9.

**Table 3 animals-13-02721-t003:** The reaction liquid of qRT-PCR.

Composition	Usage Amount (µL)
2 × ChamQ Universal SYBR qPCR Master Mix	10
PCR Forward Primer (10 µM)	0.8
PCR Reverse Primer (10 µM)	0.8
DNA template	2
RNase-free ddH_2_O	6.4

**Table 4 animals-13-02721-t004:** Effects of NaAc on blood routine indexes in postpartum dairy cows.

Item	Treatment		SEM	*p*-Value
	CON (*n* = 12)	NaAc (*n* = 12)		
**Erythrocyte System**				
Red blood cell count (RBC) # (10^12^/L)	6.24	6.24	0.22	0.984
Hematocrit (HCT) # (%)	29.09	29.17	0.98	0.935
Hemoglobin (HGB) # (g/dL)	10.09	9.83	0.28	0.356
Mean corpuscular volume (MCV) # (fL)	46.65	46.88	0.847	1.19
Mean corpuscular hemoglobin (MCH) # (pg)	16.21	15.80	0.37	0.283
Mean corpuscular hemoglobin concentration (MCHC) # (g/dL)	34.71	33.71	0.34	0.008
Red blood cell distribution width(RDW) # (%)	24.32	25.59	0.90	0.171
**Leukocyte system**				
White blood cell count (WBC) # (10^9^/L)	7.64	8.65	0.76	0.196
Neutrophili granulocyte (NEU) # (10^9^/L)	3.81	4.22	0.579	0.487
NEU (%)	49.22	48.10	3.02	0.716
Lymphocyte (LYM) # (10^9^/L)	2.70	2.89	0.208	0.368
LYM (%)	36.33	34.05	2.40	0.353
Eosinophils (EOS) # (10^9^/L)	0.15	0.24	0.104	0.346
EOS (%)	1.88	2.65	1.03	0.462
Monocyte (MONO) # (10^9^/L)	0.94	1.27	0.128	0.018
MONO (%)	12.08	14.87	1.09	0.018
Basophil (BASO) # (10^9^/L)	0.05	0.03	0.028	0.533
BASO (%)	0.52	0.32	0.27	0.469

**Table 5 animals-13-02721-t005:** Effect of diet supplemented with NaAc on the mRNA abundance of genes in PBMCs isolated from blood in postpartum dairy cows.

Item	Treatment		SEM	*p*-Value
	CON (*n* = 12)	NaAc (*n* = 12)		
Short-chain fatty acid transports				
*MCT1*	1.03	1.95	0.46	0.07
*MCT4*	1.05	2.36	0.42	0.01
*GPR41*	1.14	0.93	0.23	0.38
Oxidative stress				
*CCL2*	1.28	7.90	2.92	0.03
*NFE2L2*	1.04	2.83	0.72	0.03
*SOD2*	1.08	1.59	0.21	0.02
*NOX1*	1.13	2.54	0.68	0.05
Adhesion				
*SELL*	1.12	1.63	0.38	0.19
*ITGB2*	1.05	2.02	0.69	0.19
*ICAM1*	1.06	1.79	0.37	0.07
*TLN1*	1.06	1.12	0.23	0.80
Antimicrobial				
*MMP9*	1.14	2.51	0.77	0.09
*LAP*	1.12	2.35	0.55	0.04
*XBP1*	1.02	2.05	0.41	0.03
*TAP*	1.03	7.22	1.98	0.01
Endoplasmic reticulum stress				
*GRP78*	1.08	1.52	0.31	0.17
*ATF4*	1.07	1.05	0.17	0.92
*sXBP1*	1.15	1.30	0.29	0.61
*EIF2A*	1.08	1.45	0.26	0.16
*ASK1*	1.11	16.81	7.66	0.06
*HSP70*	1.13	2.07	0.55	0.11

**Table 6 animals-13-02721-t006:** Effect of diet supplemented with NaAc on the mRNA abundance of genes in PMNs isolated from blood in postpartum dairy cows.

Item	Treatment		SEM	*p*-Value
	CON (*n* = 12)	NaAc (*n* = 12)		
Short-chain fatty acid transports				
*MCT1*	1.99	1.50	0.92	0.60
*MCT4*	1.07	1.30	0.21	0.27
*GPR41*	3.16	1.44	2.03	0.40
Antimicrobial				
*LAP*	1.15	0.05	0.18	<0.001
*XBP1*	1.29	3.14	0.60	0.007
*MMP9*	1.15	1.45	0.47	0.52
*TAP*	1.35	0.82	0.34	0.13
Oxidative stress				
*NFE2L2*	1.07	0.89	0.22	0.41
*SOD2*	1.50	2.02	0.49	0.30
*NOX1*	1.25	0.68	0.35	0.11
*CCL2*	1.39	1.09	0.86	0.73
Apoptosis and survival				
*FAS*	1.52	1.92	0.52	0.45
*CASP2*	1.79	3.02	1.07	0.26
*BAK1*	1.76	3.70	1.25	0.14
Adhesion				
*TLN1*	1.20	2.10	0.29	0.006
*ITGB2*	1.08	3.04	0.52	0.002
*ICAM1*	1.33	1.28	0.41	0.90
*SELL*	1.51	2.92	0.59	0.02
Inflammation				
*ALOX5*	1.30	3.70	1.42	0.11
*NFKB1*	1.37	4.22	2.19	0.21

## Data Availability

Not applicable.

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
