# Peer review of "The Effects of Sodium Acetate on the Immune Functions of Peripheral Mononuclear Cells and Polymorphonuclear Granulocytes in Postpartum Dairy Cows"

_animals, 2023, doi:10.3390/ani13172721_

Round 1

Reviewer 1 Report

This manuscript mainly investigates the effects of dietary supplemented with sodium acetate on immune functions of peripheral mononuclear cells and polymorphonuclear granulocyte in postpartum dairy cows. This is a well-written paper containing interesting results because the supplementation of acetate can reduce the seizure of immune cell membranes lipids by excessive fat mobilization in postpartum period, thereby protecting immune cells. For the benefit of the reader, however, a number of points need to clarify and some statements further require justification. There are given below.

1.     In addition to grammatical and spelling errors, the introduction section should also include relevant information and appropriate applications. Please carefully check them.

2.     How do you determine the concentration of sodium acetate in this study?

3. The materials and methods lack some details, for example, how much blood collected? Does the diet used influence the results of the data and the description of isolation of peripheral mononuclear cells and polymorphonuclear granulocyte is not accurate enough, please check further.

4. In the aspect of 2.3.3: Blood Collection and Sampling Analysis, whether the indexes that you have analyzed are comprehensive?

5. Whether to consider the effect of inflammatory factors on the immune function of peripheral mononuclear cells

6. In the aspect of 4: Discussion, the discussion of each gene is not comprehensive enough, please continue to add some discussion.

7. Try to set the problem discussed in this manuscript in more clear.

   8. In discussion starting, the authors should add the some roles of acetate.

Author Response

Detailed responses can be found in the accompanying document

Reviewer 2 Report

The study aims to assess the effects of sodium acetate given as a feed additive to dairy cows post partum. Beneficial effects of acetate supplementation on dry matter intake and milk yield have been described before and hence a mechanistic study as presented here, is certainly appreciated.

The authors are invited to consider the following comments:

Simple summary:

The text contains too many unexplained abbreviations and requires English editing. Moreover, the current text does not meet the objective of a “simple summary”  (to explain the significance of the study in simples words).  Please revise.

Abstract:

Line 27: replace the word “seize” – and/or replace the entire first sentence, which is difficult to understand.

Line 29: delete: to substance

Lines 27 to 32: Should be merged or even deleted, and the abstract could start with: the aim of the current ………….

The abstract needs to provide some information on the animal trials (from M&M) before jumping into the results. Please add this information.

Line 35: How can you conclude from monocytes ratio’s (percentage) that the cows have “better immunity”  - this needs to be explained /rephrased.

Introduction:

Line: 95: Improving ER stress? Doe you mean improving the response / resilience to ER stress?

Line 95-96: How is this (correct) sentence about mastitis pathogens related he next sentences (additional explanation needed to improve readability)

M&M

Line 120: please amend:   8 mol NaAc is a concentration; please add/replace by % of dietary DM

Please explain briefly why you could not use a Latin square designed, the common experimental design in this case.  Please explain, why only a single concentration of NaAc was tested. Please add the rational for the dose selection (this should be added also to the discussion section)

Please explain why the most important plasma parameters such as acetate, glucose, BHB, and NEFA were not measured/reported. These data would allow a better comparability with data from previous experiments and would have been very relevant for the interpretation of the results.

Please note that abbreviations and specific materials need to be explained. For example: Line 167: explain ACD-A; Line 176: explain FBS

Line 184: remove one times “free”

Line 194: please give here the full names of the housekeeping genes (mentioned here for the first time)

Results and Discussion

Table3: How many samples were analysed (add n= xx to the table) Please present some additional data from method validation and reproducible experiments. For example, the repetitive analysis of one whole blood sample from one cow, and the repetitive analysis (preferentially over 3 consecutive days) of blood sample from the same animal. These validation data are of importance, as the procedure of cell isolation from peripheral blood can activate (to a varying extent) immune cells. Did you use certified LPS-free vials?

Line 211-212: You state that higher monocytes levels are indicative for better immunity. What is the rationale behind this hypothesis (monocytes also increase in number during infections!).

Lines 223-224: Likewise, you state that the upregulation of genes associated with the antimicrobial activity of PBMCs conveys protection against invading pathogens. The feeding experiment lasted for 21 days, hence one could also conclude that the upregulations of these genes is a signs of adaptation to a higher challenge with pathogens (for example inflammatory response or dysbacteriosis. This (alternative) hypothesis would be in line with the increase expression of SOD2 and other indicators of ER-stress (where the increase indicates a stress conditions, and not resilience to stressors). A similar discussion arises from the results of the qPCR analyses of PMNCs. Here also an increase in the markers of inflammation is observed, albeit statistically not significant due to an unusual high SEM value (see comment above that some validation data on reproducibility of results from native blood should be added). Please integrate these points into the discussion section.

Lines 265-267: please add a reference to this statement.

Please invite a skilled colleague to edit the article (native English speaker when possible)

Author Response

(The authors gave the same response as above.)

Round 2

Reviewer 2 Report

Thank you for addressing the specific comments in a competent manner.

The only remaining suggestions is that the manuscript would benefit from extensive English editing. Please contact the Journal's technical office for support. 

The manuscript would benefit from extensive English editing. Please contact the Journal's technical office for support. 

Author Response

I have contacted the Journal's technical office for support and the revision has been completed.